# Interacting with Members of the Public to Discuss the Impact of Food Choices on Climate Change—Experiences from Two UK Public Engagement Events

**Alana Kluczkovski [1,\*]**, **Joanne Cook [1]**, **Helen F. Downie [2]**, **Alison Fletcher [1]**,
**Lauryn McLoughlin [3]**, **Andrew Markwick [1]**, **Sarah L. Bridle [1]**, **Christian J. Reynolds [4,5]**,
**Ximena Schmidt Rivera [6]**, **Wayne Martindale [7]**, **Angelina Frankowska [1]**, **Marcio M. Moraes [8]**,
**Ali J. Birkett [9]**, **Sara Summerton [10]**, **Rosemary Green [11]**, **Joseph T. Fennell [1]**, **Pete Smith [12]**,
**John Ingram [13]**, **India Langley [14]**, **Lucy Yates [15]** and **Jade Ajagun-Brauns [1]**

1    Department of Physics and Astronomy, School of Natural Science, University of Manchester, Manchester
     M13 9PL, UK; joanne_cook.work@outlook.com (J.C.); alison.fletcher@manchester.ac.uk (A.F.);
     andrew.markwick@manchester.ac.uk (A.M.); sarah@sarahbridle.net (S.L.B.);
     angelina.frankowski@manchester.ac.uk (A.F.); joseph.fennell@manchester.ac.uk (J.T.F.);
     jade.ajaguns-brauns@student.manchester.ac.uk (J.A.-B.)
2    Department of Electrical & Electronic Engineering, School of Engineering, University of Manchester,
     Manchester M13 9PL, UK; helen.downie@manchester.ac.uk
3    National Trust, Malham Tarn Estate Office, Waterhouses, Settle BD24 9PT, UK;
     lauryn.mcloughlin@nationaltrust.org.uk
4    Department of Geography, University of Sheffield, Sheffield S10 2TN, UK; c.reynolds@sheffield.ac.uk
5    Centre for Food Policy; City, University of London, Northampton Square, London EC1V 0HB, UK
6    Institute of Energy Futures, Brunel University London, London UB8 3PH, UK; ximena.schmidt@brunel.ac.uk
7    Food Insights and Sustainability, National Centre for Food Manufacturing, University of Lincoln, Park Road,
     Holbeach PE12 7PT, UK; wmartindale@lincoln.ac.uk
8    Department of Biotechnology, Genetics and Cellular Biology, Center of Biological Sciences, State University
     of Maringá, Maringá PR 87020-900, Brazil; moraes.gen@gmail.com
9    Lancaster Environment Centre, Lancaster University, Lancaster LA1 4YQ, UK; ali.j.birkett@gmail.com
10   Department of Computer Science, School of Engineering, University of Manchester,
     Manchester M13 9PL, UK; sara.summerton@manchester.ac.uk
11   Department of Population Health, London School of Hygiene and Tropical Medicine,
     London WC1E 7HT, UK; rosemary.green@LSHTM.ac.uk
12   Institute of Biological and Environmental Sciences, University of Aberdeen, Aberdeen AB24 3UU, UK;
     pete.smith@abdn.ac.uk
13   Food Systems Transformation Programme, Environmental Change Institute, University of Oxford,
     Oxford OX1 3QY, UK; john.ingram@eci.ox.ac.uk
14   LettUs Grow, St Phillips, Bristol BS2 0QW, UK; india.langley@lettusgrow.com
15   Oxford Martin School, Oxford OX1 3BD, UK; lucy.yates@phc.ox.ac.uk
\*    Correspondence: alana.kluczkovski@manchester.ac.uk

**Abstract:** Food systems contribute to up to 37% of global greenhouse gas emissions, and emissions are increasing. Since the emissions vary greatly between different foods, citizens' choices can make a big difference to climate change. Public engagement events are opportunities to communicate these complex issues: to raise awareness about the impact of citizens' own food choices on climate change and to generate support for changes in all food system activities, the food environment and food policy. This article summarises findings from our 'Take a Bite Out of Climate Change' stand at two UK outreach activities during July 2019. We collected engagement information in three main ways: (1) individuals were invited to complete a qualitative evaluation questionnaire

comprising of four questions that gauged the person's interests, perceptions of food choices and attitudes towards climate change; (2) an online multiple-choice questionnaire asking about eating habits and awareness/concerns; and (3) a token drop voting activity where visitors answered the question: 'Do you consider greenhouse gases when choosing food?' Our results indicate whether or not people learnt about the environmental impacts of food (effectiveness), how likely they are to move towards a more climate-friendly diet (behavioural change), and how to gather information more effectively at this type of event.

**Keywords:** GHGE (greenhouse gas emissions); behaviour change; learning tools; diet; public engagement; science outreach

---

## 1. Introduction

Food systems currently constitute 21% to 37% of total human greenhouse gas emissions (GHGE), with generational and individual dietary choices influencing the magnitude of associated GHGE [1,2]. Additionally, the way in which we utilise foods to make or serve meals has a very important impact on our perception of the value of food. In order to raise awareness of what a sustainable food really is, it is important to understand how people utilise food, including food waste. The latter is a global issue, with 1.3 billion tonnes of food lost or wasted a year globally, and this is expected to grow along with population growth [3]. Hence, food waste reduction is a key aspect of sustainable food systems. In addition to awareness campaigns and intervention [4], studies have also explored how practices such as freezing foods could be used to avoid food waste and increase the value of product that could have otherwise been wasted [5].

These topics has gained public recognition, and there is an increased opportunity to develop activities to engage with educators, students, and civil society regarding sustainable food systems and in particularly about diet-related GHGE, with the aim of encouraging dietary change to mitigate climate change and promote more sustainable practices (e.g., food waste reduction). One of the types of activity is outreach, which is gaining popularity, not only because it is a communication between the researcher and civil society, but also because it is a way to present the results of research to the public [6].

It is important to engage members of the public, and in particular students, with science to increase enthusiasm for the field and inspire the next generation of scientists [7]. Furthermore, communicating scientific research content in an easy way, providing understandable language, activities and environment [8,9], has a positive benefit for the scientific community and the general public, with studies indicating that the act of engagement narrows the separation between groups [10].

Although food, diet, and nutrition are now becoming common topics for science and museum education [11–13], to the author's knowledge, there has been limited peer reviewed discussion of how to communicate (or measure the impact of communicating) the themes of food choice, climate change, sustainable diets and food waste. This article fills this gap, summarising the nature and impact of our outreach activities, describing and analysing people's interests, perceptions of and attitudes to food choices towards food and climate change. The 'Take a Bite Out of Climate Change' exhibit (hereafter 'Take a Bite') was taken to the Royal Society Summer Science Exhibition (RSSSE) in London, UK, and the Bluedot Festival (Bluedot) at Jodrell Bank Observatory in Cheshire, UK, during July 2019.

Typically, the science communication and events engagement literature focus on a broad framework of assessment [14–16], or on exhibits at specific time periods and places [17–19]. A further novelty of this article (beyond the focus on food and climate change) is (1) the multiple methods of impact assessment, and (2) multiple sites and (3) differing event durations (RSSE—a working week; Bluedot—a weekend).

The 'Take a Bite' exhibit and materials used in our engagement activities is described in the Materials and Methods section of this paper. It describes the two events and the three methods we

used for measuring impact and overall participant experience, followed by results and discussion. The last section concludes, and provides suggestions for future exhibits.

The aim of the 'Take a Bite' exhibit was to engage with the public in order to raise awareness about the impact of food choices on the climate, promote sustainable food consumption behaviours, and empower consumers with accessible knowledge to make informed decisions, as well as increasing consumer acceptance of interventions to help reduce food GHGE. In addition, the 'Take a Bite' exhibit was developed as an opportunity for individuals to engage with researchers while conveying the message that individual choices can make a difference to tackle climate change [20].

## 2. Materials and Methods

The 'Take a Bite' exhibit is a combination of several resources developed by researchers and science educators across multiple universities, charities, and commercial organisations. In this section we discuss the components of the stand, the two big events it was taken to in 2019, and the methods used to measure impact and experience.

### 2.1. The Exhibit

'Take a Bite' is an interactive exhibit, which enabled strategic researcher–learner communication (Figure 1). Scientists from relevant disciplines worked together to design and build the exhibit. The activities are designed to engage individuals in discussions regarding food production and consumption using an easy and attractive approach.

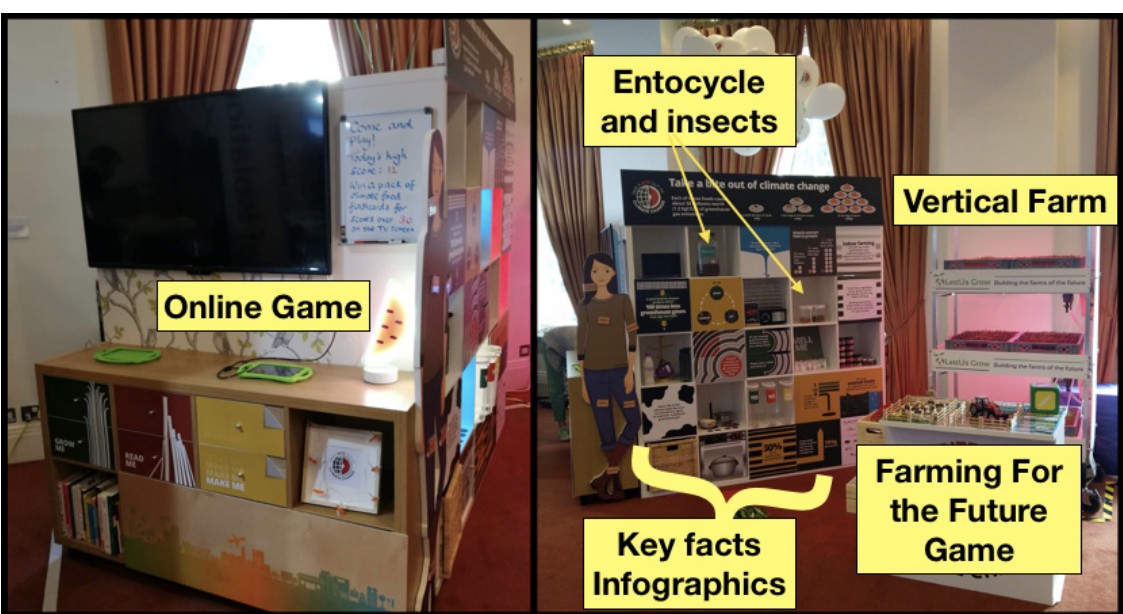

**Figure 1.** Main activities displayed in the exhibit 'Take a Bite'.

The exhibit was designed to stimulate discussion of the issues and potential opportunities for lower the GHGE of food production, such as modified agriculture and farming practices, vertical farming, manufacturing, and utilization of food waste. Similarly, the food consumption-related activities helped visitors to understand the GHGE of food items, as well as other potential low-impact and highly nutritious food (e.g., insects), and issues related to food waste at home.

The main exhibit included eight activities. The interactions started with a short conversation supported with graphic elements (e.g., balloons and fun facts), introducing the contribution of food on climate change (i.e., carbon footprint, water footprint, food waste, etc.). This short conversation aimed to highlight the importance of anthropogenic GHGE coming from food as well as to gather, in a snapshot, citizens' current knowledge, understanding and interest in the topic through question–answer and

interaction. After this initial conversation, visitors were free to walk around the exhibit to interact with other experts and activities of their choice. To conclude the interaction, a subset of visitors keen to participate were asked to answer the qualitative evaluation questionnaire (hereafter QEQ). The invitations were based on the availability of the visitor, as there were other exhibits to visit, as well as whether they accepted to participate.

Figure 2 shows the eight activities on the stand, and the route to gathering evaluation information. The visitors arrived at a random position on the stand and would sample a subset of the activities depending on their own choices. Expert communicators (ECs), drawn from across the Universities and companies involved in the development of the 'Take a Bite' exhibit, were on hand at all times to talk to the visitors about the activities.

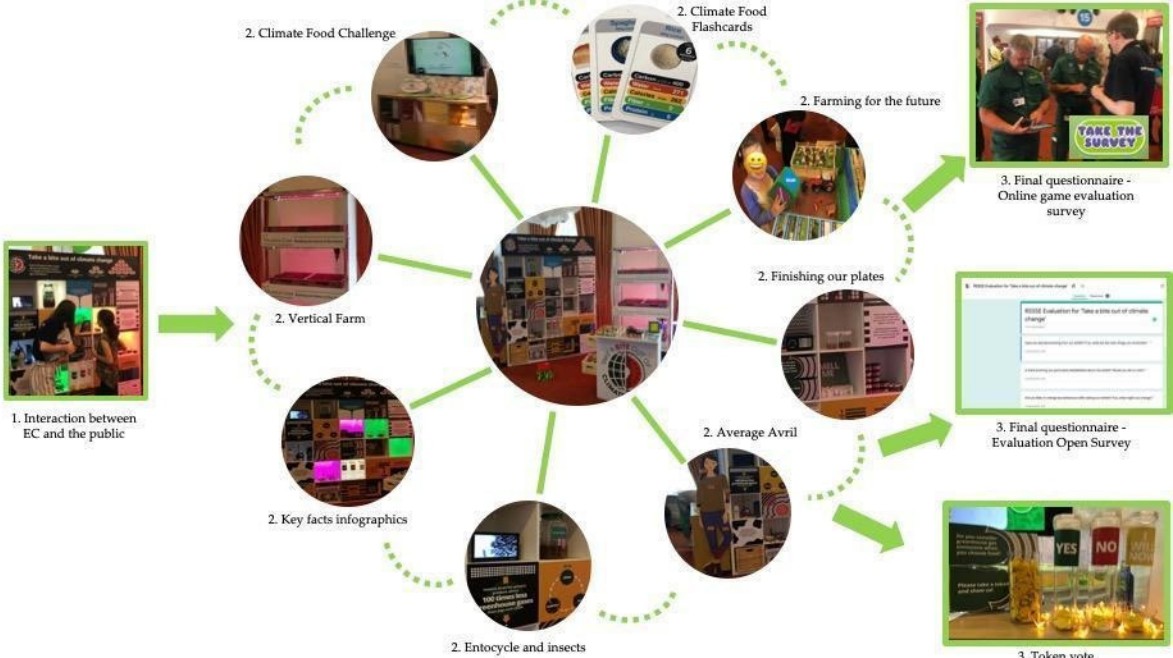

**Figure 2.** A summary of the main features of the 'Take a Bite Out of Climate Change' stand, and the three main ways we obtained evaluation information described in this article. (Figure inspired by [21]).

(1) **Infographics** were presented as a series of coloured key fact graphics (33 cm × 33 cm) on printed material with text and images, displayed on shelves as well as given as stickers. Visitors were able to read information related to the theme (e.g., "About 5% of the calories eaten by cows are burped out again as methane, a powerful greenhouse gas") if all the ECs were already busy. As speaking aids for the conversations, additional information for each topic was available as laminated A4 sheets, sourced from leading scientific publications and reports. The display infographics presented facts and images about food waste generated in the UK, carbon emissions of different food items, comparison of protein content across animal products including insects, etc. Stickers included facts and images of seasonal food—vegetables, fruits, and UK growing season (e.g., pumpkin, October–December).

(2) **Climate Food Flashcards** were made by the Greenhouse Gas and Dietary choices Open source Toolkit (GGDOT) project, which combines expertise in GHGE calculations, food nutrition and big data to create free tools to support research, communication and policy, with the goal of reducing global GHGE from food. For 'Take a Bite', GGDOT assembled a freely available spreadsheet of popular food items using typical portion sizes, with values for emissions, nutrition and water use from the scientific literature. This was built up in collaboration with academics and beyond, through a series of meetings (including hack nights). The GGDOT used these to produce the first printing (v1) of climate food flashcards which were used with visitors at the 'Take a Bite' stand. The flashcards (v1) are a set of 56 cards that each show a serving of a specific food. Each flashcard comprised an Open Access

image of the food, as well as GHGE, nutrition and water footprint information corresponding to the serving size. Some foods were included twice with different transport systems (air vs. land/rail/water transport) to clearly illustrate the impacts of both food and transportation. The carbon footprint (total GHGE produced across the life- cycle of a product or service) was represented in two different ways: (i) grams of $CO_2e$ and (ii) equivalent number of minutes of driving a car. The cards also showed the water footprint (litres; total freshwater used to produce the food) and nutritional information (protein in g; calories in kCal; fibre in g). The flashcards were used to engage with participants to highlight differences between food impacts and nutrition and impacts of production and transport systems. The ECs encouraged participants to play and make their own games, for instance "the challenge game" where each participant turns over a card and the lowest of a player-selected category (e.g., 'protein') wins the round.

(3) **Climate Food Challenge** is an online game developed for the 'Take a Bite' project. It was played on either an iPad tethered to the stand, or (via a QR code) on the participant's own phone, tablet, or laptop. It takes data regarding portion size and emissions from the GGDOT spreadsheet (see above) for 28 foods. The game asks participants to rank 3 portions of food in order of carbon footprint (e.g., $gCO_2e$), from lowest to highest. Each participant was asked to order as many combinations as they could in one minute—if the participant got multiple correct combinations, they were given bonus points and extra time. Though random, the first triplet shown to participants typically contained one 'low' carbon footprint food (e.g., lettuce 48 $gCO_2e$), and two 'higher' carbon footprint foods—e.g., beef and sausages (1939 $gCO_2e$, 1035 $gCO_2e$, respectively). As the game progresses, the ratio of differences in carbon footprint between foods became smaller and smaller, increasing the difficulty. At the end of each game, there was an opportunity to quit, play again, or take a survey about their experience.

This game was also taken to the National Video Gaming Museum, Sheffield in November 2019, for an Economic and Social Research Council (ESRC) Festival of Social Sciences event. The impact feedback of this event is included in the online Supplementary Information (SI).

(4) **Farming for the Future** is an interactive board game. The goal is to open up a discussion on (environmentally) sustainable farming and discuss each element of the possible improvements. A model farm is set out in a traditional mixed farm layout with toy animals (i.e., cows, sheep) on a board representing a farm. Participants role an oversized dice to decide which category they need to improve (out of GHGE, soil health, energy efficiency, biodiversity, water use efficiency or economic performance). Participants can then choose from nine improvement actions, including: (i) planting trees, (ii) installing renewable energies—a wind turbine, (iii) stop ploughing (going no-till), (iv) using GLADDIS (a 'state-of-the-art' mobile trace gas and stable isotope tracking laboratory developed by the University of Manchester) to measure GHGE, (v) reducing animal numbers, (vi) stopping the use of fertiliser and pesticides, (vii) use precision agriculture, (viii) add beehives and/or (ix) swap fences for hedgerows. Participants choose a maximum of 3 interventions and arrange them on the farm. Each intervention was given a score based on how well it solves the given category (e.g., water use efficiency), so a total score for each player can be calculated. The EC then discusses the best options for farm environmental sustainability.

(5) **A Vertical Farm** module (a multi-tier aeroponic indoor farming kit containing microgreens) was provided by LettUsGrow. This display allowed participants to learn by experiencing (e.g., seeing, touching, smelling) indoor farming, and talk about its potential role in reducing emissions, transport and land use. ECs also asked participants to guess what was growing in the module, and this then led into discussion about the nutrition of microgreens and other plants that can be grown using indoor farming.

(6) **Entocycle and Insect Protein Displays** aim to show innovative initiatives which use insects to provide low-impact and efficient protein sources for human and animal consumption, and to increase the economic value of food waste. At the Royal Society Summer Science Exhibition, this was shown as a sealed box display containing live black soldier fly larvae consuming waste from the brewing industry. Information was displayed about lifecycle and GHGE credentials of the insects. Insect cookbooks

were also provided on a bookshelf of the exhibit. These displays led ECs to open conversations about the issue of quality feedstocks for feeding animals, and better use of food waste. Additionally, based on the fact that GHGEs from aquaculture and chicken depend significantly on feed, ECs could also discuss issues about where fish come from, modern aquaculture practices, amongst others. At the Bluedot Festival, we offered a variety of edible insects from the company Crunchy Critters as well as pink marshmallows (containing cochineal), and we thus presented a selection of commercially farmed insects that had been purged, cleansed, dried and packed safely to be suitable for human consumption and that provide high protein and fibre content, low carbohydrates, and minerals and essential amino acids. Similarly, the pink marshmallows open a discussion on how insects have been used by industry as red food colourings—the carmine pigment from cochineal gives a widely used pink colouring, therefore many of us will have eaten insects already without knowing it.

(7) **"Finishing our plates"** was an 'unwasted food sampling' sensory experience. The ECs led participants through a discussion on food loss and waste throughout the food system (including on-farm production, harvest, manufacturing and at home), as well as food processing technologies (preservation, drying, milling, etc.) Fruit and vegetables were used as examples of highly nutritious foods that are easily lost/wasted due to perishability (e.g., soft fruit) and unattractiveness of some parts (e.g., cauliflower stalks). A live display of the fruit (e.g., strawberries) dehydration unit was provided, while samples of fruit and vegetable flours and smells were captured in sample tubes. This approach allowed ECs to discuss the production of high-end value products from highly perishable food or avoidable food losses as well as alternative uses for fresh produce to conserve flavour and nutrition. WRAP's Love Food Hate Waste food waste reduction flyers and giveaways (bag clips and pasta measurers) were also used at this activity.

(8) **Average Avril** is a life-size 2D cartoon of the "average human" on the planet. Avril's body symbolises the contribution to climate change of daily life activities. The body sections are clearly identified with a percentage corresponding to the six largest contributors: food (25%), thermal comfort (18%), industry and travel (15% each), washing (11%) and waste (6%), numbers based on Bojana et al. [22] The ECs guided participants to decide where to place, on Avril's body, the magnets that represented the six main categories of daily activities, with the aim of allowing visitors to have time to consider the impacts of their food choices in the context of other daily activities.

## 2.2. The Two Events

We ran the 'Take a Bite' stand at two main events (i) the Royal Society Summer Science Exhibition 2019 (RSSSE) and (ii) the Bluedot Festival 2019 (Bluedot). The RSSSE is a free annual event attended by members of the general public and groups of school students, running for 7 consecutive days at the beginning of July, in London, UK. Bluedot is a family-friendly annual music and science festival attended by paying members of the public, running for 3 consecutive days at the end of July.

Both events generally received a varied audience including families, students, teachers and members of the public who were interested in science. However, the two events are different in venue and purpose. At the RSSSE, individuals could see some information about the content of the stand in an event booklet and choose the stands they wanted to visit. Since Bluedot is also a music festival, many of the visitors might not have any background in science, nor an intent to visit and engage with the stand. The ECs took care to not supervise people while they were voting. At Bluedot, the token voting session was relocated to a top shelf to make the access easier.

The RSSSE has reported the following rough visitor numbers: around 1700 students and teachers in school groups, and over 9750 public visitors. Bluedot was attended by approximately 21,000 people over the weekend of the music festival [23].

## 2.3. Measuring Impact and Experience

We assessed volume of interactions by counting the number of visitors (adults, children and teenagers) actively engaged (i.e., in conversation with an EC) with the exhibit hourly, with a 10–30-min

window. At the RSSSE we counted it on three exhibit days: Friday, Saturday and Sunday, and at Bluedot we counted it on two exhibit days: Saturday and Sunday. These numbers were used to estimate visitor numbers throughout the week based on Friday as a standard weekday, taking into account different opening hours throughout the week at the RSSSE. We did not count interactions on Wednesday and Thursday evening due to the reduced number of ECs on the stand; therefore, these times have not been included in the data presented here.

We used three interactive methods to measure participant experience. (1) a structured, qualitative evaluation questionnaire (QEQ) was conducted face-to-face by one of the ECs specifically responsible for collecting the survey, with randomly selected consenting participants (see details in Appendix A). To analyse the impact of the activities with the public, it was asked which activity respondents liked and disliked and whether they learnt something after visiting the exhibit (question 1 and 2); (2) an online multiple-choice survey (OMS) was deployed on tablets, available at the end of playing the climate food challenge game (CFC) (see details in Appendix B); and (3) a token drop voting activity, in which a token was given to 920 participants, who were then asked to 'drop the token' or 'vote' accordingly to their reply to: "Do you consider greenhouse gases when choosing food?". Three containers were provided with the following answers: "yes", "no", or "I will now". It is worth noting that this activity was not easily accessible to most visitors at the RSSSE because the exhibit was often crowded with people, not easy to see or access to take a vote. Also, visitors had to be asked to vote rather than finding it themselves even when the exhibit was quiet, possibly because it was not clear where to get a token from.

Finally, interactions on social media through engagement on Twitter® were counted, counting how many times the particular Twitter user engaged with our content (clicks anywhere on the tweet, including retweets, replies, follows, likes, links, cards, hashtags, embedded media, username, profile photo, or tweet expansion). Details are given in Appendix C.

## 3. Results and Discussion

This section presents the assessment and discussion of the engagement numbers, and the three interactive evaluation mechanisms used to measure people's interests, perceptions and attitudes related to food choices and their impacts on climate change. First, Section 3.1 shows a brief summary of the visitors attending the events—RSSSE and Bluedot. Then, Sections 3.2 and 3.3 assess the outcomes of the QEQ and the OMS, respectively. Finally, Section 3.4 presents the outcomes of the token drop voting activity analysis.

### 3.1. Visitor Numbers

The estimated total number of visitors at the RSSSE was therefore 6287 and at Bluedot was 581 (Figure 3a,b). The 'Take a Bite' stand engaged with approximately 6868 people in these two events, 64% adults (20 years old and older), 16.8% teenagers (ages 13–19) and 18% children (under 13 years old). The results for the counting number showed that Sunday was the busiest day at the RSSSE, with the highest number of adults and children visiting. However, the number of teenage visitors was higher during each weekday than during each weekend day, as expected due to secondary school group visits taking place from Monday to Friday. Consequently, it is important to consider which days an event will take place, so that it reaches the largest number of people in the target audience. Furthermore, comparing the total number of visitors (RSSSE 11450, Bluedot 21,000 people), this topic has proven to be of interest to a large audience, as well as people from a variety of age groups.

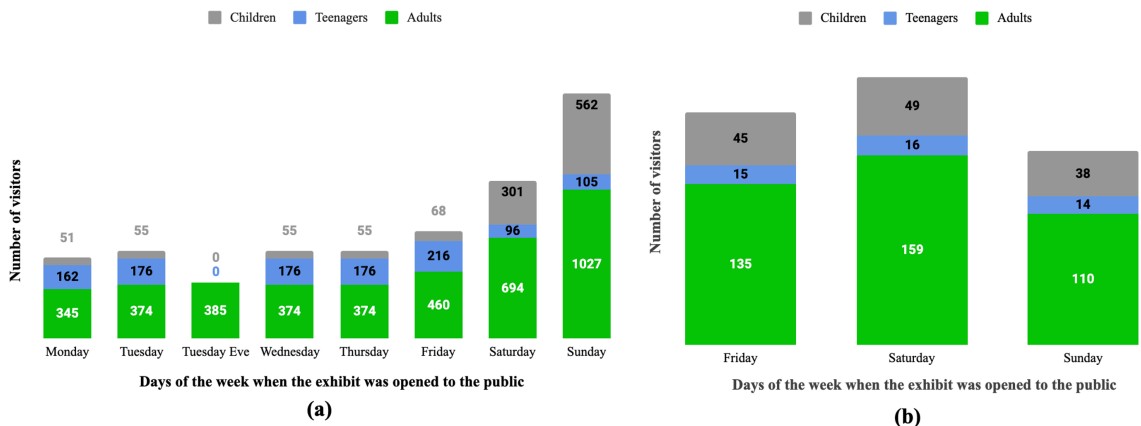

**Figure 3.** Estimated number of visitors who engaged with 'Take a Bite Out of Climate Change' during (**a**) the Royal Society Summer Science Exhibition and (**b**) Bluedot Festival.

### 3.2. Qualitative Evaluation Questionnaire (QEQ)

In total, 78 responses were collected from the QEQ. Thirty-seven visitors answered the questionnaire at the RSSSE (47%), and 41 at the Bluedot Festival (52%). The QEQs show primarily that people were interested in the topic presented. The results also showed that 1% of the visitors answered the QEQ, which should be improved in future events, in order to have substantial data and therefore be more accurate.

Figure 4 shows what visitors learnt from the exhibit as well as the most remembered things. The topics most often mentioned at RSSSE (carbon footprint of food (32.4%) and aeroponics (21.6%)) were different to those mentioned at Bluedot (eating seasonal and/or local (33%) and food choices and environmental impact, approximately 19%). This might have occurred due to different audiences at the events and/or the fact that the QEQs did not reach a more homogeneous sample. Thus, it is important to consider the correct activity to develop the target audience, as well as cover a larger number of responses, when using this survey. Surprisingly, themes less commonly mentioned at Bluedot were insect protein and aeroponics (even though they were prominently displayed on the stand at Bluedot), both with only 3.3% of the responses.

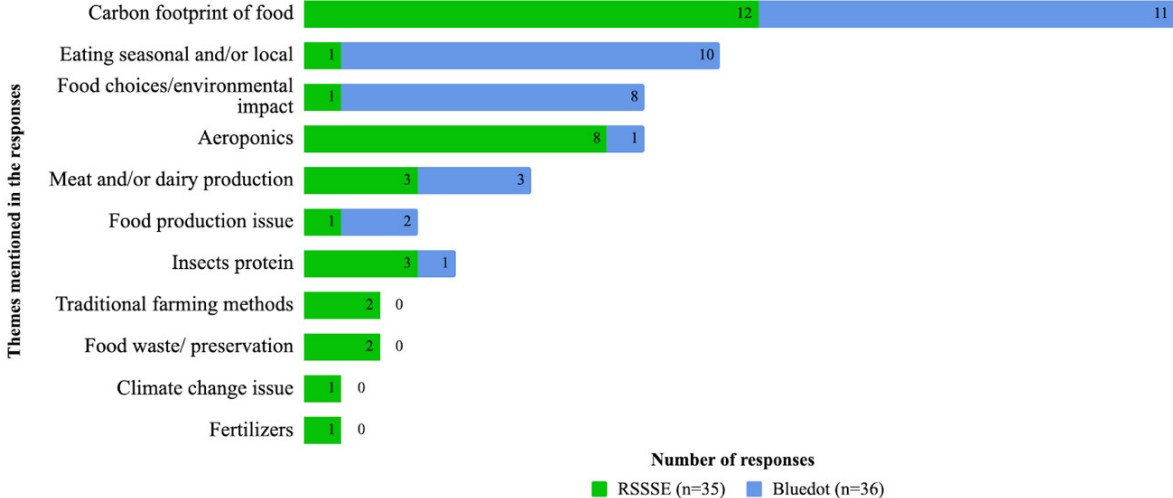

**Figure 4.** What visitors say they have learnt from the activities at the RSSSE and Bluedot Festival, based on the qualitative evaluation questionnaire (QEQ). Seventy-one responses were collected (35 for RSSSE and 36 for Bluedot). It should be noted that 5.4% of the individuals did not answer this question at the RSSSE, and 16.7% did not answer at Bluedot.

All the respondents said they had learnt something at the exhibit, as well as having liked everything about the exhibit (RSSSE 19%, Bluedot more than 29%). The majority of participants liked the ECs. They were remembered by 19% of the visitors that answered the questionnaire at the RSSSE and by 12.9% at Bluedot. The ECs had an extremely important role, developing interesting conversations about the theme and attracting people to see the stand. Furthermore, planning the event in advance and providing training to the ECs was fundamental to ensure they were all talking, not only about relevant topics, but also exemplifying these topics with scientific and accurate information. In addition, it was possible to observe that some activities were remembered more than others, for example climate food flashcards and key facts infographics. Climate food flashcards obtained the most responses at Bluedot (14 visitors liked it), while three mentioned it as preferred in the RSSSE. These interactive methodologies to impact the public about food choices and environment proved to be more efficient. The next most well remembered activities were 'farm for the future', 'food waste/preservation', and 'climate food challenge'. We note that some people interacted with several activities but did not mention when answering the survey, or they did not engage with all the activities in the stall. These activities could be improved further for future exhibits. In addition, two responses at the RSSSE mentioned that the exhibit was too busy and there was too much information (5%). This may indicate that the use of many activities in a single event can affect the public's perception and learning, since it may not be possible to participate in all activities. Due to a lack of information, or answers not related to the question, it was not possible to define 5% of the responses at RSSSE and 9.8% at Bluedot, which could be improved by using a qualitative survey.

Question number three asked whether respondents are likely to change any behaviour after seeing the exhibit. Most participants said they were keen to change behaviour regarding food choices after visiting the stand (24 at the RSSSE and 27 at Bluedot), which might indicate a positive result for the activities developed. Following the above responses, individuals stated what behaviour they would change (Figure 5). In both events two main themes were observed, (1) eating food with a lower carbon footprint and (2) reducing meat consumption. Activities such as climate food flashcards and the climate food challenge are directly related to these answers, as they discuss reductions on red meat consumption, as well as the carbon footprint of different foods, and they can be used in future events to engage with people. Some of the themes were mentioned only in one of the two events, such as 'research more about food' at Bluedot, and 'already on a carbon-friendly diet, but keen to improve' at the RSSSE. This variation might have occurred due to the reduced number of questionnaires collected. It is suggested that we increase the number of responses in future events.

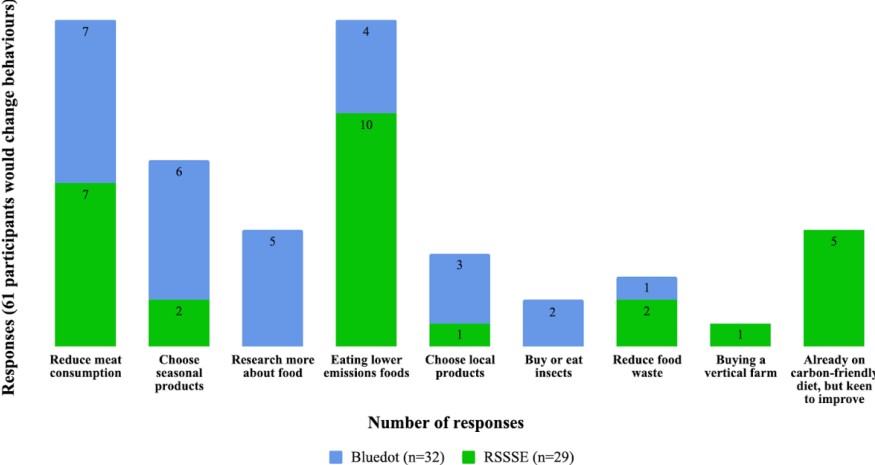

**Figure 5.** Habits that behaviour respondents might change after seeing the exhibits. For this analysis only positive responses were considered ('yes' and 'maybe') from question 3 of the QEQ. Sixty-one responses were obtained. Both events obtained five responses not related to the question, or where it was not possible to identify the content.

Figure 6 shows which tools respondents would find useful to help them make low greenhouse gas emission-based food choices. The majority of responses were positive at both events (39 at Bluedot and 31 at the RSSSE, making up 95% and 84% of the totals respectively). Only 20 participants said they would not have an interest in tools to help them make more climate-friendly food choices. Analysing the type of learning tools people said they would like, across both surveys, the majority of answers, approximately 50%, were related to a tool, whether it is online or not. For instance, online tools include apps, online games, online calculators and websites.

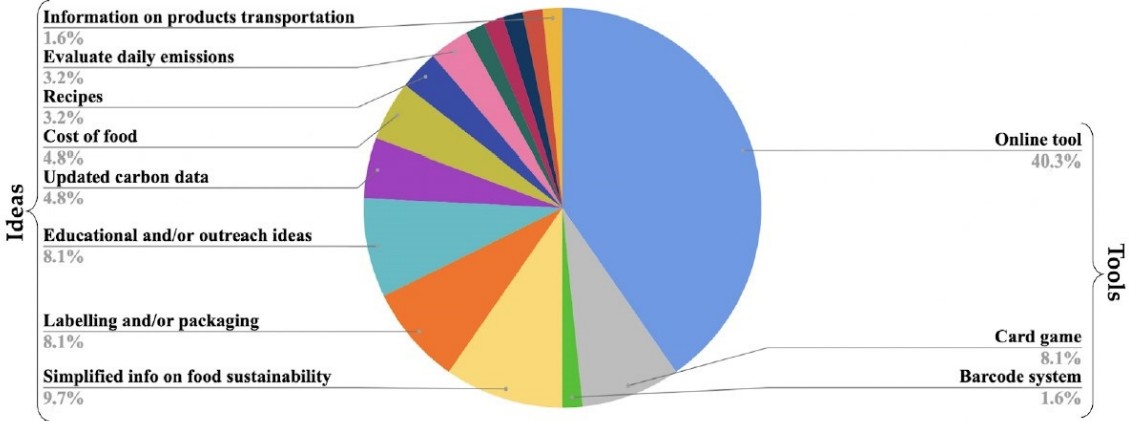

**Figure 6.** Types of learning tools respondents would like to use in the future. It shows the percentage of what participants would like to have as a learning tool, and ideas to help them make more friendly food choices in both exhibits. n = 84; undefined responses (n = 8) and questions not answered (n = 13) are not shown in the chart.

A variety of ideas were also presented by the participants, some of which could be used by policymakers, such as changes in labelling and/or packaging, some of which could be developed by universities, such as updated carbon data. It shows that people are interested in making changes to their diet, and they only need information and tools to make it easier.

When asked which tools participants would like to have to help them making more friendly food choices, most of them mentioned the development of an app or other online tool, such as a website and online carbon footprint calculator. Further action towards this result was setting up an event (September 2019) to engage with members of the public, to co-develop ideas for tools and projects that would help reduce greenhouse gas emissions from food.

### 3.3. Online Multiple-Choice Survey (OMS)

In total, 312 people responded to the OMS after playing the climate food challenge (CFC). Question 1 of the OMS survey asked the age of respondents. It was possible to find only an approximate link between the age categories used by the Office of National Statistics and our survey, as the categories used did not overlap exactly. Overall, 42% of people who completed our survey were young people (<25 years) and 59% were 25+, whereas the UK population consists of 30% young people, and 70% of people are 25+ [24]. To determine whether the population sampled is representative of the UK, information on gender, ethnicity, sexuality, religion and socioeconomic status should be gathered.

Question two asked participants to indicate their current diet. The diet categories in question two (located in Appendix B) were further grouped into: vegan, vegetarian, pescatarian, flexitarian and meat eaters. Diet 1 (in the response options to question two in Appendix B) was vegan, diets 2, 3, 4 and 5 were vegetarian, diet 6 was pescatarian, diets 7 and 8 were flexitarian and diets 9 and 10 were meat eaters. It was found that flexitarian was the most popular diet, with 54.4% of people indicating they followed this diet. To simplify question two, a matrix style question where individuals can indicate how often they eat specific food items (e.g., red meat, milk, tofu, eggs) should be used. This would

simplify the survey by allowing question two to be grouped with questions four and five, which ask individuals how often they eat specified types of protein (questions four and five can be found in Appendix B).

Figure 7 shows people's diets, along with their concern for their environmental impact, before playing the CFC game. Initially, it seemed as though people who mostly/only eat products derived from plants are the most aware/concerned with their environmental impact before playing the game. However, people who mostly/only eat products derived from plants made up 4.5%, 95% CI (2.3%, 6.7%) of survey respondents. The number of respondents is not large enough to draw any meaningful conclusions about associations between food preferences and understanding of environmental pressures.

| Q10a Q2 | Don't know | Not at all concerned | Slightly concerned | Concerned | Extremely concerned |
|---|---|---|---|---|---|
| I frequently eat meat and I am not interested in trying vegetarian food | 1 | 4 | 6 | 3 | 0 |
| I often eat meat and I occasionally eat vegetarian food | 6 | 7 | 14 | 19 | 1 |
| I often eat both meat and vegetarian food | 5 | 6 | 31 | 58 | 6 |
| I mostly eat vegetarian food and occasionally eat meat | 3 | 3 | 9 | 33 | 14 |
| I eat fish dairy and eggs in addition to products derived from plants | 1 | 2 | 4 | 14 | 7 |
| I eat dairy and eggs in addition to products derived from plants | 0 | 1 | 2 | 6 | 7 |
| I eat dairy in addition to products derived from plants | 0 | 0 | 0 | 1 | 0 |
| I eat eggs in addition to products derived from plants | 0 | 1 | 0 | 3 | 0 |
| I mostly only eat products derived from plants | 0 | 1 | 1 | 8 | 9 |
| I only eat products derived from plants | 0 | 2 | 1 | 6 | 6 |

**Figure 7.** A colour-coded table that shows the relation between the type of diet people say they follow, and their level of concern for the environment. There are 312 survey responses illustrated in the figure. Vegans made up 4.5%, 95% CI (2.3%, 6.7%) of respondents, vegetarians 13.2%, 95% CI (9.4%, 17%), pescatarians 9.2%, 95% CI (6.1%, 12.3%), flexitarians 54.4%, 95% CI (47.71%, 61.1%) and meat eaters 18.8%, 95% CI (15.4%, 22.2%) of respondents.

The responses to questions three and four provided specific information regarding people's diets and why they choose to follow it. The three dominant reasons people gave as to why they followed their diet were: environmental concerns, health concerns and animal welfare concerns, which were also the first three response options for the question. In future surveys, randomising the answer options should eliminate bias towards selecting the first answer options. Questions five and six were long and confusing. Therefore, analysis of these questions was neglected.

Question seven asked individuals how likely they were to switch to a climate-friendly diet in the next 12 months. In total, 79.3%, 95% CI (71.0%, 87.6%), indicated that they were at least 'somewhat likely' to switch to a climate-friendly diet in the next 12 months. This question assumed individuals were aware of the meaning of the term 'climate-friendly diet' and understood what types of diets are friendlier to the environment. Future surveys should include a clarification of the term 'climate-friendly diet'.

The results of question eight illustrated that 63.8%, 95% CI (59.1%, 67.7%) of people who are considering switching their diet, would do so for environmental reasons. The results of question eight also showed that 50.0%, 95% CI (33.7%, 66.3%), of people who already followed a climate-friendly diet, did so for health reasons.

The results for question eight suggest that people want to be more environmentally friendly with their food choices, but due to the ambiguity of 'environmental concerns', it cannot be concluded that this is because of food GHGE. Modifying the answer options for 'environmental concerns' to 'plastic concerns' and 'greenhouse gas emissions concerns' will remove this ambiguity. It will also enable inferences to be drawn about whether the game has encouraged dietary change by increasing awareness/concern of GHGE. Furthermore, it was shown that individuals who would change their diet for environmental reasons have the opinion that climate friendlier diets are healthier. This opinion is consistent with that given by Nelson et al. [25,26] To understand the extent of any potential behavioural change, a follow-up survey 3, 6 or 12 months after the exhibit will be considered.

Figure 8 illustrated the reasons why people were not likely to switch to a climate-friendly diet. Given the high uncertainty illustrated by the confidence intervals in Figure 8, it was not possible to determine the main reason why people did not want to switch to a climate-friendly diet. If the dominant reasons were identified, they could be used to develop tools to encourage dietary change. For example, if individuals do not want to follow a climate-friendly diet due to health concerns, then healthy, climate-friendly recipes could be developed.

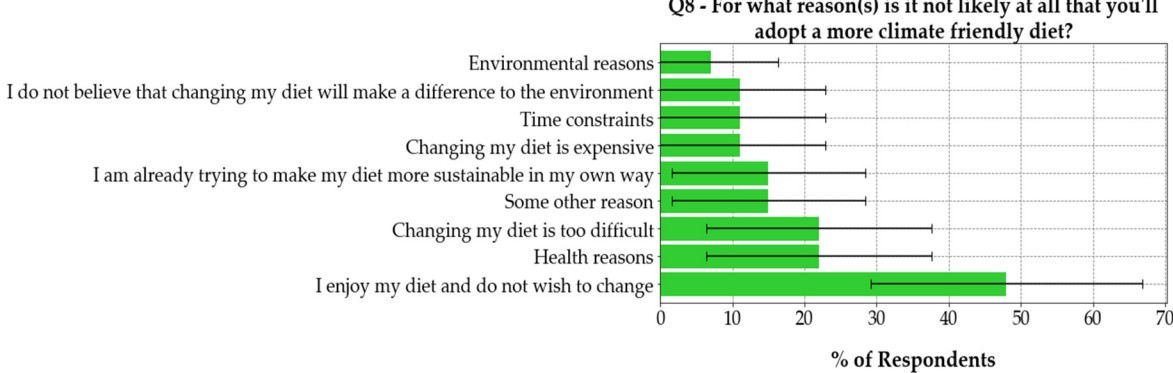

**Figure 8.** Reasons why respondents are not likely to adopt a more climate-friendly diet. In question seven of the multiple-choice online survey, 27 respondents indicated that they were 'not very likely' or 'not at all likely' to change their diet. In question eight, this subset of respondents were asked why they were not likely to change their diet—these results are shown in the figure. In the figure, 95% confidence intervals are shown.

Figure 9 shows the results of questions 9a, 9b, 10a and 10b of the OMS survey. These questions aimed to gauge the impact of the game on individuals' awareness and concerns about food greenhouse gas emissions. The percentage of individuals who were 'extremely aware' of the environmental impact of their food choices increased from 14.3%, 95% CI (10.4%, 17.9%) to 43.0%, 95% CI (37.4%, 48.6%). The percentage of individuals who were 'extremely concerned' increased from 16.2%, 95% CI (12.1%, 20.3%) to 36.3%, 95% CI (31.0%, 41.6%). The percentages of individuals who were 'extremely aware' of and 'extremely concerned' about the environmental impact associated with their food choices increased

after playing the game. Furthermore, there was a greater increase in individuals' awareness of their environmental impact than their concern.

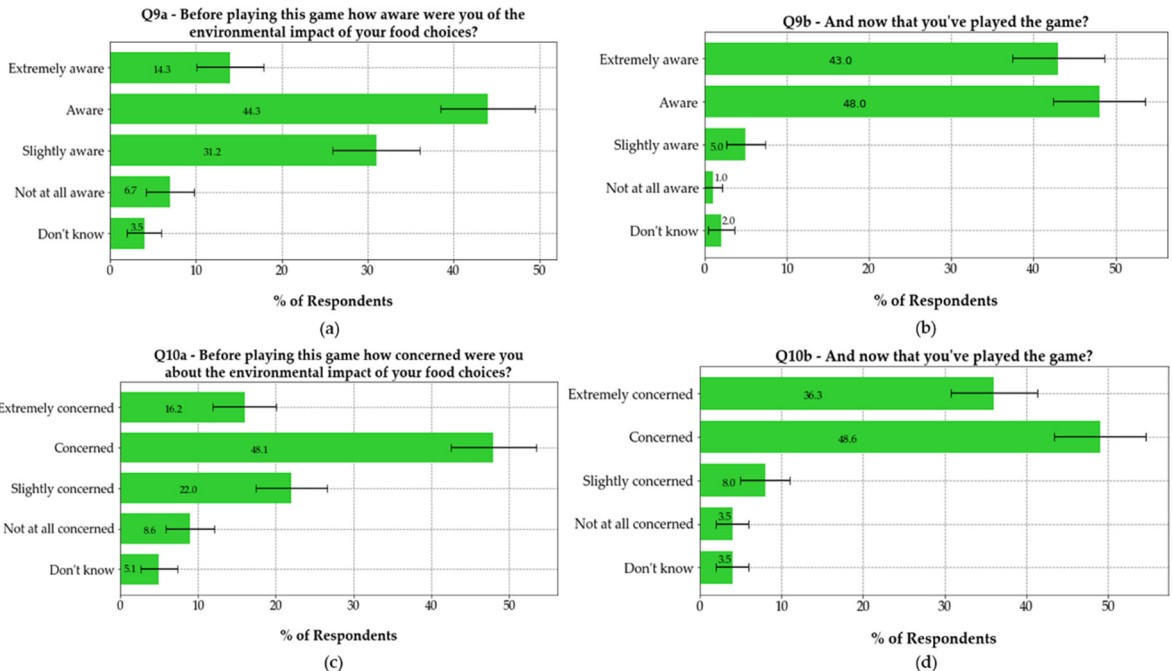

**Figure 9.** Levels of awareness (upper) and concern (lower) about environmental impacts of people's food choices before (left) and after (right) playing the online game. The bars and values written on the bars indicate the percentage of people who selected a particular response. In the figure, 95% confidence intervals are shown.

The term 'environmental impact', which appears in questions 9 and 10, is ambiguous. Individuals may interpret 'environmental impact' as referring to plastic packaging, food greenhouse gas emissions, or something else entirely. Furthermore, given that 81% of people who completed the survey did so at either the RSSSE or Bluedot Festival, it is likely that they had conversed with a staff member before playing the game to discuss food greenhouse gas emissions. These facts together mean that question 9a and 10a are biased towards people having a greater understanding of food greenhouse gas emissions before playing the game.

The results for the OMS survey show that the game is a good tool for communicating food GHGE, however, it is not as good at communicating the negative impacts of greenhouse gas emissions (i.e., people gained an idea of the scale of individual foods' greenhouse gas emissions, but the game did not provide a link to the environmental consequences of their diet). To counter this, it is suggested that a future version of the game colour-codes the amount of greenhouse gas emissions associated with each food using a colour-coded system similar to the one known as the food traffic light labelling system [27], which is already familiar to consumers in the UK. It shows on the front of the pack whether a product is high (red), medium (amber) or low (green) in macronutrients such as calories, proteins, etc. [28] For example, if steak, milk and apple were to appear together, when the answer appears on the screen, steak could be labelled in red, milk in orange and apple in green. This would help the player understand the extent of the environmental impact. Additionally, it is suggested that an introduction to the game could describe what greenhouse gas emissions are, and how they impact the environment. This may help increase individuals' concern for their food greenhouse gas emissions, and in turn, could prove a valuable tool in encouraging dietary change.

### 3.4. Token Drop Voting Activity

Figure 10 shows the analysis on the token drop voting session at both events. In total 920 random visitors voted; 448 visitors answered the question at the RSSSE, and 472 at the Bluedot Festival.

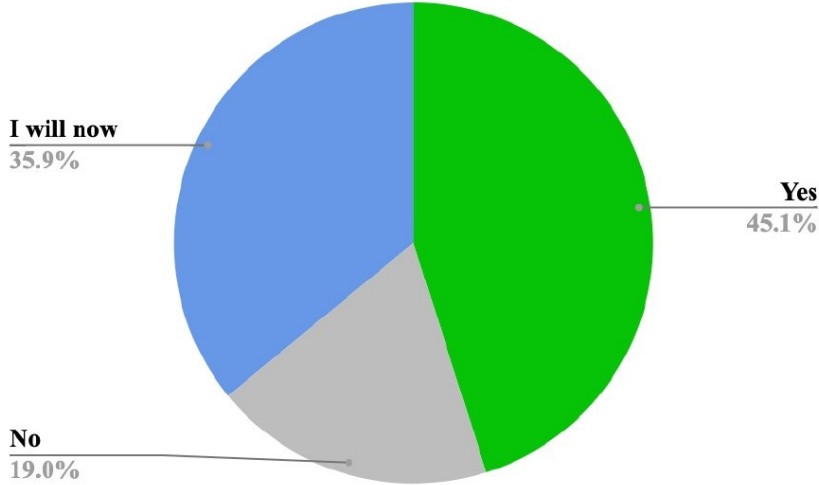

**Figure 10.** Token drop voting responses to evaluation question "Do you consider greenhouse gases when choosing food?" at Bluedot and the RSSSE (n = 920 responses).

The majority of the visitors (45.1% at the RSSSE and 55% at Bluedot) said they consider GHG emissions when choosing food. About 36% said that they will consider greenhouse gases in their daily food choices after visiting the RSSSE exhibit. This pattern was not observed at Bluedot, where 21% did not consider greenhouse gases when choosing food. Only 19% of the visitors did not consider GHGE when choosing food, while more than 23% of the visitors said they will consider GHG when choosing food.

### 3.5. Social Media Interactions

Twitter® was used to advertise the 'Take a Bite' stand to a wider audience using social media. The 'Take a Bite' stand had 86 tweets on Twitter® and 159,334 people viewed the tweets (see details in Appendix C). The number of engagements (retweets, shares, comments and likes) was 2237, whilst the number of clicks on the exhibit tweets was 221. In total 173 Twitter® users retweeted content about the exhibit.

### 3.6. Confirmation between Methods and Events

In general, participants of the QEQ said they planned to change their behaviour regarding sustainable food choices. This was observed in question 3 of the open survey as well as on the token drop voting activity, which shows that 723 participants would change something in their behaviour on food choices. The behaviours most mentioned were to reduce meat consumption and eating fewer foods with high carbon footprints, followed by choosing seasonal and local products. Both events together obtained a similar number of responses related to the carbon footprint of food. At Bluedot there were more responses about eating seasonal and/or local food choices, and environmental impact (10 and 8, respectively) compared with the responses at the RSSSE (one for eating seasonal and/or local and one for food choices and environmental impact). Aeroponics and insect's protein were themes more remembered at the RSSSE than Bluedot. The topic 'meat and/or dairy' was equally mentioned at both events (three responses in each event). Food production as an issue had almost the same number of responses in both exhibits. Traditional farming methods, food waste and preservation, climate change as an issue and fertilizers were themes only mentioned by visitors at the RSSSE (note that the food waste parts of the stand were not taken to Bluedot).

The results of this study are of importance in showing whether people learnt about the environmental impacts of food (effectiveness), which was shown to be positive. Furthermore, it was possible to analyse how likely they were to follow a climate-friendly diet in the future. For instance, most people said they want to make changes in their diet to help the environment and their health. Thus, we presented some strategies for researchers who want to develop outreach activities, such as the 'Take a Bite' exhibit, which help to discuss relevant issues related to climate change.

## 4. Conclusions and Advice for Future Public Engagement Events

The three data collection methods have shown that the 'Take a Bite' exhibit has influenced at least 723 participants to change their behaviour and food choices. The 'Take a Bite' stand can be seen as a success, with longer term follow-up surveys (3, 6 or 12 months after the exhibit) suggested to monitor longer term fact recall and impact (participant behaviour change). Below, we highlight notable findings and provide suggestions for future food and climate related public engagement and science communication activities:

- We found that all the QEQ participants said they had learnt something in the exhibit as well as having liked something about the activities. This can be attributed to the wide variety of activities and engagement options on the stall. Of all our exhibit features, the most popular were found to be the climate food flashcards and key facts infographics.
- Although the 'Take a Bite' exhibit had a strong web presence (www.takeabite.info) with content, games, and additional material, nearly 50% of people surveyed (QEQ) had requests for the development of further online tools. Future exhibits would be wise to continue this policy of developing all exhibit features to also be available online. Likewise, future exhibits (and app developers) should engage with this appetite for tools and content related to food and climate change.
- The expert communicators (ECs) were remembered by 19% of the visitors (QEQ). We highlight the important role of the ECs as being crucial for the success of the events, developing interesting conversations about the theme and attracting people to see the stand. We encourage all future exhibits to have a wide team of (well rostered) ECs to draw on.
- Our review of our participants' feedback has found that the climate food challenge online game, though popular and a good tool for communicating food greenhouse gas emissions, was not as good a tool for communicating the negative impact that greenhouse gas emissions have. Future exhibits need to consider that games and activities need to show (1) the scale and greenhouse gas emissions impact of different dietary changes (as the climate food challenge did), as well as (2) communicate to participants what reductions to emissions would actually achieve (i.e., the possible consequences of dietary change). Although these two actions do not have to occur in the same game or activity, the lack of this second action was a weakness of the 'Take a Bite' exhibit, as identified through our analysis.
- The words and terms used in participation surveys are important. Future exhibits need to take into account the broad scope of definitions and concepts when communicating or measuring concerns and behaviour change, with phrases such as 'environmental concerns', 'environmental impact' or 'climate-friendly diet' open to ambiguity and assumption. Future surveys should include an explanation of the meaning of specific terms to prevent participants being confused by the wording of the question. In addition, there were complaints of too many answer options in a 'short' survey, and this led to participant confusion. To address this, we suggest future exhibits change from a multiple-choice survey to a matrix style question to resolve this issue.

**Author Contributions:** Conceptualization and development of exhibit, S.L.B., H.F.D., A.F. (Alison Fletcher), L.M., A.M., C.J.R., X.S.R., W.M., A.F. (Angelina Frankowska), A.J.B., S.S., R.G., J.T.F., P.S., J.I., I.L. L.Y.; methodology, software and formal analysis, A.M., A.K., J.C.; M.M.M.; writing—original draft preparation and visualisation, A.K., J.C., H.F.D.; writing—review and editing, A.K., S.L.B., C.J.R., X.S.R., A.F. (Alison Fletcher), H.F.D., J.A.-B.; project administration, A.F. (Alison Fletcher), H.F.D.; the Farming for the Future game was developed by L.M. and H.F.D. The climate food challenge game was taken to the National Video Gaming Museum, Sheffield, by C.J.R. and A.M. All authors have read and agreed to the published version of the manuscript.

**Funding:** This research and public engagement activity was funded through multiple research grants. N8 Agrifood funded projects "Greenhouse Gas and Dietary choices Open-source Toolkit (GGDOT) hacknights" and, with the University of Manchester, funded the development of the climate food flashcards. Additional funding was provided by the HEFCE Catalyst-funded N8 AgriFood Resilience Programme and matched funding from the N8 group of Universities, and the STFC Food Network+. Development of the "Take a Bite out of Climate Change" stand and the "Climate Food Challenge" video game, as well as attendance at the Royal Society Summer Science Exhibition and the Bluedot Festival in July of 2019, was supported by funding from STFC Food Network+ and the HEFCE Catalyst-funded N8 AgriFood Resilience Programme, matched funding from the N8 group of Universities and additional funding from the University of Manchester. This project arose from the N8 AgriFood-funded project "Greenhouse Gas and Dietary choices Open-source Toolkit (GGDOT) hacknights.' Part of this work was supported by the Wellcome Institutional Strategic Support Fund award [204796/Z/16/Z]. We are grateful for the funding from the Wellcome Trust Manchester Institutional Strategic Support Fund, the STFC Food Network+, N8 Agrifood and the University of Manchester. During the organisation of this research, the running of the events and the writing of this paper, Sarah Bridle and Christian Reynolds were supported in-part though the STFC GCRF funded project "Trends in greenhouse gas emissions from Brazilian foods using GGDOT" (ST/S003320/1). Christian Reynolds received additional funding from NERC to support an Innovation Placement at the Waste and Resources Action Programme (WRAP) (Grant Ref: NE/R007160/1). Alana Kluczkovski was supported through a University of Manchester GCRF Fellowship funded through the University of Manchester internal Research England GCRF QR Fund. Ximena Schmidt Rivera was supported through Brunel University internal Research England GCRF QR Fund.

**Acknowledgments:** Thanks to collaborating researchers from major UK programmes such as Global Food Security Programme, Met Office, Scottish Climate Change Centre of Expertise (ClimateXChange), Society for the Environment, industry, and universities across the UK. The GGDOT project is grateful to the National Diet and Nutrition Survey for the nutrition data used in the climate food flashcards and to the Water Footprint Network for the water use numbers. We are grateful to the Wellcome Trust-funded Livestock, Environment and People (LEAP) project for printing and showcasing a preliminary version of the climate food flashcards at their conference in 2018. We thank the 'Take a Bite Out of Climate Change' Advisory Board for advice and ideas that shaped the exhibit. We are grateful to the 'Take a Bite Out of Climate Change' expert communicators team for engaging with the public and helping to get survey responses at the Royal Society Summer Science Exhibition 2019 and the Bluedot 2019 Festival. We give thanks to the following members of the 'Take a Bite Out of Climate Change' team: Peter Wooton-Beard, Flora Hetherington, Taff Morgan, Aled Jones, Edward Pope, Geoff McBride, Dave Johnson, Sally Howlett, Laurence Stamford, Charles Veys, Neil Chalmers, Rachel Marshall, India Langley, Matt Holl, Katherine Denby, Lucy Yates, Marianna Ventouratou-Morys, Maia Elliott, James Stockdale, Sonal Choudhary, Ilias Kyriazakis, Duncan Harding, Claire Hoolohan, Mark Reed, Bruce Grieve, and Emma Wilcox. We are grateful to Rachael Hand for help developing the proposal and ideas for the exhibit. The climate food challenge was developed by Andrew Marwick (University of Manchester) in collaboration with Strangely Retro Games, using a subset of the GHGE developed for the climate food flashcards.

**Conflicts of Interest:** The authors declare no conflict of interest.

## Appendix A. Qualitative Evaluation Questionnaire (QEQ)—Long-Form Evaluation Survey

After visiting the exhibit, selected random individuals were invited to complete a survey, comprising four open questions that gauged the person's interests and perceptions of food choices and attitudes towards climate change. The aim of this survey was to evaluate the impact of the proposed activities in the stand. At the RSSSE the responses were collected via Google Docs Form (using tablets) typed by the visitors and sent online. At the Bluedot Festival, questionnaires were printed, and visitors were able to handwrite their responses, and after the event the responses were typed in a Google docs Form and the database analyzed as follows.

The questions were:

1. Have you learned anything from our exhibit? If so, what are the main things you remember?
2. Is there anything you particularly liked/disliked about the exhibit? Would you tell us more?
3. Are you likely to change any behaviours after seeing our exhibit? If so, what might you change?

4. Would you find it useful to have any tools to help you make low greenhouse gas emission's food choices? If so, what would you want from them?'

It is worth noting that, in general, the subjects presented in the responses were mentioned by an expert communicator engaging with a visitor.

**Appendix B. Online Multiple-choice Survey (OMS)—Multiple Choice Survey**

1. **How old are you?**

Under 18
18–24
25–34
35–44
45–54
55–64
65+

2. **How would you describe your current diet?**

I frequently eat meat and I am not interested in trying vegetarian food
I often eat meat and I occasionally eat vegetarian food
I often eat both meat and vegetarian food
I mostly eat vegetarian food and occasionally eat meat
I eat fish dairy and eggs in addition to products derived from plants
I eat dairy and eggs in addition to products derived from plants
I eat dairy in addition to products derived from plants
I eat eggs in addition to products derived from plants
I mostly only eat products derived from plants
I only eat products derived from plants

3. **For what reasons do you currently choose to follow this diet? (Tick all that apply)**

Environmental concerns
Animal welfare concerns
Health concerns
Cost
Availability of food
Religious beliefs
Other

4. **How often do you eat beef or lamb?**

Never
Rarely
Once a month
Once a week
2–3 days a week
4–5 days a week
Every day
Don't know

5. **How often do you choose lower-impact protein options (such as chicken, veggie, sausages, quorn, tofu) over higher impact options (lamb or beef) to reduce your environmental impact?**

Never
Rarely
Once a month
Once a week
2–3 days a week
4–5 days a week
Every day
Don't know

6. **As a result of playing this game, how likely are you to consider lower impact protein options over higher impact protein options in the future to reduce your environmental impact?**

Very likely
Likely
Somewhat likely
Not very likely
Not likely at all
I already always choose low impact options

7. **In the next 12 months, how likely are you to adopt a more climate friendly diet?**

Very likely
Likely
Somewhat likely
Not very likely
Not likely at all
I already always choose low impact options

8. **For what reason(s) would you consider adopting a more climate friendly diet? (Tick all that apply)**

Religious reasons
Health reasons
Environmental reasons
Some other reason

**For what reason(s) is it not likely at all that you'll adopt a more climate friendly diet? (Tick all that apply)**

Religious reasons
Health reasons
Environmental reasons
I enjoy my diet and do not wish to change
I do not believe that changing my diet will make a difference to the environment
Changing my diet is too difficult
Changing my diet is expensive
I'm already trying to make my diet more sustainable in my own way
Time constraints
I will in the future

Some other reason

**For what reason(s) did you adopt a more climate friendly diet? (Tick all that apply)**

Religious reasons
Health reasons
Environmental reasons
Some other reason

**9. (a) Before playing this game how aware were you of the environmental impact of your food choices?**

Not at all aware
Slightly aware
Don't know
Aware
Extremely aware

**(b) And now that you've played the game?**

Not at all aware
Slightly aware
Don't know
Aware
Extremely aware

**10. (a) Before playing this game how concerned were you about the environmental impact of your food choices?**

Not at all concerned
Slightly concerned
Don't know
Concerned
Extremely concerned

**(b) And now that you've played the game?**

Not at all
Slightly concerned
Don't know
Concerned
Extremely concerned

Participants for the survey were selected using convenience sampling at the 'Take a Bite out of Climate Change' stand at the RSSSE and Bluedot festival. The option to take the survey appeared after playing the CFC game on an iPad. Some individuals were playing the game unsupervised and will have clicked the 'Take a survey' button spontaneously. Usually, a member of staff asked the person if they were willing to complete it, to encourage more responses. Not every person who played the game was asked to complete the survey because this depended on how busy the stand was, and whether the staff members felt comfortable asking them to complete the survey. Additionally, a link to the game was available on the https://takeabite.info/ webpage and the www.ggdot.org webpage shared and tweeted.

In total there have been 2572 plays of the game in the period 01/07/2019–05/08/2019. The results of the survey analysis will be valid for the people who played the CFC game up to 05/08/2019. However,

they should not be expected to represent how the population of the UK would respond after playing the game. There was a strong selection effect as 81.4% of the sample were people attending the RSSSE and Bluedot festival. Furthermore, the people electing to visit the stand may have been particularly interested in the topic of food and/ or climate change. Of those visiting only a fraction chose to play the game, and of those, only a fraction completed the survey (Table A1). These people are more likely to be interested in science than the population of the UK. Hence, the sample was not representative of the UK population.

**Table A1.** The number of times the CFC game was played and the number of survey responses for each event (including non-response surveys). Excluding non-response surveys, there were 356 surveys with at least one question answered.

| Date | Event | Number of Games Played | Number of Survey Responses |
|---|---|---|---|
| 01/07/2019–07/07/2019 | RSSSE | 2097 | 271 |
| 19/07/2019–21/07/2019 | Bluedot | 283 | 45 |
| 08/07/2019–18/07/2019 22/07/2019–05/08/2019 | Other | 192 | 72 |

## Appendix C. Social Media Evaluation

Table A2 shows the engagement through Twitter®over the course of the RSSSE and Bluedot. In total, the 'Take a bite out of climate change' stand had 86 tweets. The number of visualizations, however, was much more expressive (159334). The number of engagements was 2237, whist the number of clicks in the exhibit tweets was 221. In total 173 Twitter® users retweet content about the exhibit.

**Table A2.** Number of interactions on social media. It shows the number of online interactions using Twitter® The take a bite out of climate change had in both events.

| Number of Tweets | Impressions * | Engagements ** | Link Clicks | Retweets |
|---|---|---|---|---|
| 86 | 15.9334 | 2237 | 221 | 173 |

* Number of times a user saw the tweet on twitter. ** Engagement on Twitter® means how many times the particular Twitter user engaged with your content: clicks anywhere on the tweet, including retweets, replies, follows, likes, links, cards, hashtags, embedded media, username, profile photo, or tweet expansion.

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
