# Peer review of "Interacting with Members of the Public to Discuss the Impact of Food Choices on Climate Change—Experiences from Two UK Public Engagement Events"

_sustainability, doi:10.3390/su12062323_

Round 1
Reviewer 1 Report
This manuscript deals with an issue within the WHO agenda of health topics deserving further scientific attention. The study is comprehensive, well done and structured. Based on the results, the authors highlight the possibility/necessity of the other researches related to “the impact of food choices on climate change” idea. About the paper itself, some minor modifications are required, in my opinion, in order to facilitate the understanding of the information presented.
Line 69. I suggest writing what GHGE means immediately after it is first used, as the abbreviation is used directly in the Keywords section, and what it means appears only further below in the text.
Line 72. “…contribute with approximately a…”
Line 75. “…to develop activities to engage with educators, students…”
Lines 92-93. Unfinished phrase.
Lines 100-101. Unclear. Please rephrase (there are two words “typically” and “specific” repeated twice).
Lines 104. Before Section 2, there is no mention about Section 1. Should consider rephrasing and better structuring.
I suggest ending the Introduction section with the aim of the research (lines 85-91).
Lines 141-142. Please rephrase.
In the Results and discussion section, the results are very well structured and presented, in a descriptive manner, but as a reader, I need to feel/read as well as an interpretative author's manner.
Lines 589-594. I suggest rephrasing and making some amendments from a grammatical and punctuation point of view.
Author Response
We are grateful to the editor for facilitating the review and to the reviewers for their comments, all of which have been addressed as detailed below.
Line 69. I suggest writing what GHGE means immediately after it is first used, as the abbreviation is used directly in the Keywords section, and what it means appears only further below in the text.
We acknowledged the reviewer’s comment and added the meaning.
Line 72. “…contribute with approximately a…”
We acknowledged the reviewer’s comment and added text. See changes in line 73.
Line 75. “…to develop activities to engage with educators, students…”
We acknowledged the reviewer’s comment, however it was not clear what the issue was. We have reviewed and improved the phrase. See changes in lines 84-86.
Lines 92-93. Unfinished phrase.
We considered the reviewer’s comment and finished the phrase. See changes in lines 94-97.
Lines 100-101. Unclear. Please rephrase (there are two words “typically” and “specific” repeated twice).
We acknowledged the reviewer’s comment and improved the text. See changes in lines 103-104.
Lines 104. Before Section 2, there is no mention about Section 1. Should consider rephrasing and better structuring.
We considered the reviewer’s comment and improved the section. See changes in lines 108-111.
I suggest ending the Introduction section with the aim of the research (lines 85-91).
We considered the reviewer’s comment and improved the introduction including aim of the research. See changes in lines 112-117.
Lines 141-142. Please rephrase.
We acknowledged the reviewer’s comment and improved the writing in lines 143-144.
In the Results and discussion section, the results are very well structured and presented, in a descriptive manner, but as a reader, I need to feel/read as well as an interpretative author's manner.
We considered the reviewer’s comment and improved the results and discussion section making it more interpretative. Changes were made in the entire section.
Lines 589-594. I suggest rephrasing and making some amendments from a grammatical and punctuation point of view.
We considered the reviewer’s comment and improved the text. See changes in lines 544-549.
Reviewer 2 Report
Review of the manuscript ‘interacting with members of the public to discuss the impact of food choice on climate change – experiences from two UK public engagement event’
Dear authors,
This manuscript present a very interesting case study of implementing an outreach/information activity for the general public to discuss the impact of our food choices on the environment. The manuscript is good in its present form, I only recommended some small changes to the order and detail of some of the parts.
Line 76: ‘food GHG emission’, I would consider to rephrase this to diet-related GHGE.
Line 81: main aim of this activity to me would seem to inform and help to induce change in behavior in those people, not primarily to increase enthusiasm and inspire a new generation of new scientist?
Line 83: Very vague what the authors mean by ‘narrowing the separation between groups’? Usually giving information is better understand by higher educated people thus increasing the separation even more.
Line 103: The novelty of multiple timepoints seems a bit too strong, given that both events are within the same month.
Line 108-109: section is out of place, probably an erroneous enter.
Line 132: add ‘of’ between GHGE – food production.
Line 142: ‘subset of visitors’. I would recommend to include a more critical review on the ‘random’ selection to invite someone or not to participate. How does this potentially affect the observed responses? When it was busy for example, you probably occupied by questions and assisting rather than asking people to fill in some forms.
Line 248-249: please include the name of the reference within the sentence.
In de result section some parts are double but just differently phrased, please remove:
Line 303-305 is a double of line 302 Line 436-437 is a double of 433-435 Line 453-455 is double of line 448-450Line 381: what is meant by ‘validated’?
Line 425: maybe give some examples of responses that lack some information
Line 438-439: You say the age groups are not comparable to the national statistics, (<25,>25) but you have more information on specific age ranges (appendix) so these you should be able to link approximately?
Line 442: adding info, so leave out age with you did include.
Personally, I feel that the list of co-authors is quite extensive for a study design like this.
Author Response
We are grateful to the editor for facilitating the review and to the reviewers for their comments, all of which have been addressed as detailed below.
Line 76: ‘food GHG emission’, I would consider to rephrase this to diet-related GHGE.
We considered the reviewer’s comment and rephrased the term. See changes in line 85.
Line 81: main aim of this activity to me would seem to inform and help to induce change in behavior in those people, not primarily to increase enthusiasm and inspire a new generation of new scientist?
We acknowledged the reviewer’s comment. However, it seems to be a misunderstanding between the aim of this communication and the one of the exhibit. In lines 97-99, the aim of this manuscript is described as follows: “This article fills this gap, summarising the nature and impact of our outreach activities, describing and analysing people’s interests, perceptions and attitudes of food choices towards food and climate change”. Line 112, describes the aim of the activity: “The aim of the ‘Take a Bite’ exhibit was to engage with the public in order to raise awareness about the impact of food choices on the climate, promote sustainable food consumption behaviours and empower consumers with accessible knowledge to make informed decisions, as well as increasing consumer acceptance of interventions to help reduce food GHGE”. We improved the structure of the introduction, making clear the aim of the communication/paper.
Line 83: Very vague what the authors mean by ‘narrowing the separation between groups’? Usually giving information is better understand by higher educated people thus increasing the separation even more. I would say that communicating scientific research content in an easy way narrows the separation between group?
We appreciated the reviewer’s comment and added the following changes to clarify the sentence “narrowing the separation between groups”: ‘Furthermore, communicating scientific research content in an easy way, providing understandable language, activities and environment [8,9] has a positive benefit for the scientific community and the general public on, with studies indicating that the act of engagement narrows the separation between groups [10]’. See lines 90-94 to see changes.
Line 103: The novelty of multiple timepoints seems a bit too strong, given that both events are within the same month.
We are grateful for the comment. We have changed the text and added the following “1) the multiple methods of impact assessment, 2) multiple sites and 3) differing event durations (RSSSE- a working week; Bluedot- a weekend)”. See changes in lines 105-108.
Line 108-109: section is out of place, probably an erroneous enter.
We considered the reviewer’s comment and improved the text allocating the section in a correct place. See changes in lines 75-78.
Line 132: add ‘of’ between GHGE – food production.
We acknowledged the reviewer’s comment and added the preposition. See changes in line 135.
Line 142: ‘subset of visitors’. I would recommend to include a more critical review on the ‘random’ selection to invite someone or not to participate. How does this potentially affect the observed responses? When it was busy for example, you probably occupied by questions and assisting rather than asking people to fill in some forms.
We appreciated the reviewer’s comment and added the following explanation in lines 145-147: “To conclude the interaction, a subset of visitors keen to participate were asked to answer the Qualitative Evaluation Questionnaire (hereafter QEQ). The invitations were based on the availability of the visitor, as there were other exhibits to visit, as well as whether they accepted to fill the survey”. Additionally, we have improved section 2.3, explaining the selection and surveying processes. The following sentence was added: "We used three interactive methods to measure participant experience: (1) a structured, Qualitative Evaluation Questionnaire (QEQ) was conducted face-to-face by one of the ECs specifically responsible for collecting the survey, with randomly-selected consenting participants (see details in Appendix A)". See changes in lines 284-287.
Line 248-249: please include the name of the reference within the sentence.
We acknowledged the reviewer’s comment and included the reference as follows: “The body sections, clearly identified with a percentage corresponding to the six largest contributors: food (25%), thermal comfort (18%), industry and travel (15% each), washing (11%) and waste (6%), numbers based on Bojana et al [22]”. See changes in line 254.
In de result section some parts are double but just differently phrased, please remove:
Line 303-305 is a double of line 302
We considered the reviewer’s comment and improved the text removing double sentences. See changes in lines 312-313.
Line 436-437 is a double of 433-435
We considered the reviewer’s comment and improved the text removing double sentences. See changes in lines 407-411.
Line 453-455 is double of line 448-450
We considered the reviewer’s comment and improved the text removing double sentences. See changes in lines 414-418.
Line 381: what is meant by ‘validated’?
We acknowledged the reviewer’s comment. ‘Validated’ was used to explain that it was not used all the questionnaires collected due to some reasons, such as presenting only email address. We rephrased the sentence and added it at the caption of Figure 5 in lines 379-380.
Line 425: maybe give some examples of responses that lack some information
We acknowledged the reviewer’s comment. An example of a response that lack some information is ‘med science’. We re-structured the results and discussion section making it more interpretative removing this sentence.
Line 438-439: You say the age groups are not comparable to the national statistics, (<25,>25) but you have more information on specific age ranges (appendix) so these you should be able to link approximately?
We considered the reviewer’s comment and improved the text, adding the following information: ‘It was possible to find only an approximate link between the age categories used by the Office of National Statistics and our survey as the categories used did not overlap exactly’. See changes in lines 407-409.
Line 442: adding info, so leave out age with you did include.
We considered the reviewer’s comment and removed age. See changes in line 412.
Personally, I feel that the list of co-authors is quite extensive for a study design like this.
We appreciated the reviewer’s concern. However, this work would not have been possible without the contribution of all of the authors of this manuscript. The contribution of each author has been detailed and acknowledged in section Author Contributions, as requested in the Journal’s guidelines.
Reviewer 3 Report
General Comments
This is an interesting study that shares the experiences on interactions with pubic about impacts of different food choices on climate change. Mainly, the authors capture outcomes to two public events in UK, where the authors exhibit “Take a Bite out of Climate Change” stand. This exhibit consists of three components to make public aware on impacts of different food choices on environment and to get insights on their willingness to change their diets toward climate friendly one. Although this is an interesting manuscript that contributes in the field of science communication, the manuscript needs to be improved. I would suggest addressing the following comments to improve the presentation of the manuscript.
Mainly, “Results and Discussion” section consists of description of method as well. It would make more sense to include method used in method section instead of mixing them with results. Additionally, this section could be shortened. Currently, it consists of a detailed description of the figures. It would have need better to highlight the main results instead of providing a repeated description of the figures in the text. There is also repetition of some text in this section. Please read the manuscript and avoid the repetitions. In general, results need better presentation. It is also confusing that the manuscript jumps from results of one event to another event in inconsistent way.
Specific Comments
L72-74: Please also have a look on the IPCC SRCCL Chapter 5 for the GHG emissions from food systems.
L100-103: The flow of this paragraph needs to be improved.
L103: “paper” instead of “communication”
L108-116: Yes true, however, these statements here do not match the flow of the manuscript.
L112-113: Statement regarding food waste is doubtable. There are several studies and initiatives to reduce food waste (e.g. https://www.save-food.org/)
L432-436: Where can read see these results?
L436-437: This is repetition of the previous lines (L432-436).
L453-457: Repetition of L444-452.
L469: This figure is not based on correlation analysis. Thus, it would better to avoid such specific term as “correlation”.
L477: What about questions 3-6? A better framing of the result section would improve the manuscript.
L488-492: Yes true, however, please also highlight finding of this study.
L499-503: This is confusing. L444 says question 2 is about diet they current follow.
L509: Regarding traffic light labelling, please also consider that different color has different meaning in different culture. Red is not bad in all culture.
L525: Above it says OMS survey instead of CFC survey.
Author Response
We are grateful to the editor for facilitating the review and to the reviewers for their comments, all of which have been addressed as detailed below.
Mainly, “Results and Discussion” section consists of description of method as well. It would make more sense to include method used in method section instead of mixing them with results. Additionally, this section could be shortened. Currently, it consists of a detailed description of the figures. It would have need better to highlight the main results instead of providing a repeated description of the figures in the text. There is also repetition of some text in this section. Please read the manuscript and avoid the repetitions. In general, results need better presentation. It is also confusing that the manuscript jumps from results of one event to another event in inconsistent way.
We considered the reviewer’s comment and improved the results and discussion section, making it more interpretative and moving methods used from this section to the Materials and Methods. Changes were made in the entire section, considering both events consistently in order to improve the presentation. We improved the text removing double sentences.
Specific Comments
L72-74: Please also have a look on the IPCC SRCCL Chapter 5 for the GHG emissions from food systems.
We considered the reviewer’s comment and improved the text, added IPCC SRCCL report as a reference and re-wrote the following sentence: ‘Food systems currently contribute with 21 to 37% of total human greenhouse gas emissions (GHGE), with generational and individual dietary choices influencing the magnitude of associated GHGE [1,2]’. See changes in lines 73-75.
L100-103: The flow of this paragraph needs to be improved.
We considered the reviewer’s comment and improved the text. See changes in lines 104-108.
L103: “paper” instead of “communication”
We considered the reviewer’s comment and replace communication for an article. See changes in line 106.
L108-116: Yes true, however, these statements here do not match the flow of the manuscript.
We considered the reviewer’s comment and improved the text allocating the section in a correct place. See changes in lines 75-82.
L112-113: Statement regarding food waste is doubtable. There are several studies and initiatives to reduce food waste (e.g. https://www.save-food.org/)
We considered the reviewer’s comment. We realised that this sentence might lead to wrong conclusions. Hence, the sentence was erased. See changes in lines 75-80.
L432-436: Where can read see these results?
We acknowledged the reviewer’s comment. However, we opted to present the most relevant images to aid the reader's understanding and engagement. The total number of participants to the survey along with question 1 was presented as a text description as we considered easier to understand without adding a figure. We improved the structure of this section, making it clearer.
L436-437: This is repetition of the previous lines (L432-436).
We considered the reviewer’s comment and improved the text removing double sentences. See changes in lines 407-411.
L453-457: Repetition of L444-452.
We considered the reviewer’s comment and improved the text removing double sentences. See changes in lines 414-418.
L469: This figure is not based on correlation analysis. Thus, it would better to avoid such specific term as “correlation”.
We considered the reviewer’s comment. We realised that this sentence might lead to wrong conclusions. Hence, the word was changed to ‘relation’. See changes in line 431.
L477: What about questions 3-6? A better framing of the result section would improve the manuscript.
We considered the reviewer’s comment and improved the results and discussion section making it more interpretative. We added the following sentence to presented results of questions 3-6: ‘The responses to questions 3 and 4 provided specific information regarding people’s diets and why they choose to follow it. The three dominant reasons people gave as to why they followed their diet were: environmental concerns, health concerns and animal welfare concerns which were also the first three response options for the question. In future surveys randomising the answer options should eliminate bias towards selecting the first answer options’. See changes in lines 437-441.
L488-492: Yes true, however, please also highlight finding of this study.
We considered the reviewer’s comment and improved the text. See changes in lines 457-460.
L499-503: This is confusing. L444 says question 2 is about diet they current follow.
We considered the reviewer’s comment. We realised that this section was out of place. Hence, the sentence was erased. See changes in lines 460-461.
L509: Regarding traffic light labelling, please also consider that different color has different meaning in different culture. Red is not bad in all culture.
We considered the reviewer’s comment and improved the text as follows: ‘To counter this, it is suggested that a future version of the game colour codes the amount of greenhouse gas emissions associated with each food using a colour coded system similar to that known as food traffic light labelling system [27], which is already familiar to consumers in the UK. It shows on the front-of-pack whether a product has high (red), medium (amber) or low (green) in macronutrients such as calories, proteins, etc [28]. See changes in lines 498-502.
L525: Above it says OMS survey instead of CFC survey.
We considered the reviewer’s comment and corrected the word. See changes in line 495.
Round 2
Reviewer 3 Report
General Comments
I thank the authors for addressing almost all the comments. The manuscript has improved. However, before considering for publication, I suggest considering provided specific comments below. Additionally, there is potential to shorten the manuscript, mainly in result and discussion section, by avoiding a detailed description of the figures.
Specific Comments
L54: Emission from food system is more than a quarter of global greenhouse gas emissions
L75: Reference 2 should be the IPCC SRCCL Chapter 5 instead of the IPCC SR15
L82: Please consider to mention that food waste is increasing across the world (Hic et al 2016, ES&T).
L84-88: When food waste is introduced above, it would be better to mention here as well instead of focusing only on diet changes. This also holds for 95-103.
Round 2
We are grateful to the editor for facilitating the review and to the reviewer for the comments, all of which have been addressed as detailed below.
Reviewer #1:
I thank the authors for addressing almost all the comments. The manuscript has improved. However, before considering for publication, I suggest considering provided specific comments below. Additionally, there is potential to shorten the manuscript, mainly in result and discussion section, by avoiding a detailed description of the figures.
We considered the reviewer’s comment and improved the structure of the results and discussion section, making it more interpretative according to their suggestion on the first review round. Changes were made in the entire section, considering both events consistently in order to improve the presentation. However, we consider that further changes will compromise the content and understanding of the paper.
L54: Emission from food system is more than a quarter of global greenhouse gas emissions
We acknowledged the reviewer’s comment and improved the text. See changes in line 54.
L75: Reference 2 should be the IPCC SRCCL Chapter 5 instead of the IPCC SR15
We acknowledged the reviewer’s comment and corrected the text. The correct reference is in line 646 as follows: Mbow, C., C. Rosenzweig, L.G. Barioni, T.G. Benton, M. Herrero, M. Krishnapillai, E. Liwenga, P. Pradhan, M.G. Rivera-Ferre, T. Sapkota, F.N. Tubiello, Y. Xu, 2019: Food Security. In: Climate Change and Land: an IPCC special report on climate change, desertification, land degradation, sustainable land management, food security, and greenhouse gas fluxes in terrestrial ecosystems [P.R. Shukla, J. Skea, E. Calvo Buendia, V. Masson-Delmotte, H.-O. Pörtner, D.C. Roberts, P. Zhai, R. Slade, S. Connors, R. van Diemen, M. Ferrat, E. Haughey, S. Luz, S. Neogi, M. Pathak, J. Petzold, J. Portugal Pereira, P. Vyas, E. Huntley, K. Kissick, M. Belkacemi, J. Malley, (eds.)]. In press.
L82: Please consider to mention that food waste is increasing across the world (Hic et al 2016, ES&T).
We acknowledged the reviewer’s comment and added the information in a citation number 3. See changes in lines 77-79.
L84-88: When food waste is introduced above, it would be better to mention here as well instead of focusing only on diet changes. This also holds for 95-103.
We acknowledged the reviewer’s comment and improved the structure of the paragraph. See changes in lines 83-86 and 95-98.